# Transcriptomic Identification of Immune-Related Hubs as Candidate Predictor Biomarkers of Therapeutic Response in Psoriasis

**DOI:** 10.3390/ijms26178118

**Published:** 2025-08-22

**Authors:** Elisabet Cantó, María Elena del Prado, Eva Vilarrasa, Anna López-Ferrer, Francisco Javier García Latasa de Araníbar, Maria Angels Ortiz, Marta Gut, Maria Mulet, Anna Esteve-Codina, Ruben Osuna-Gómez, Albert Guinart-Cuadra, Luís Puig, Silvia Vidal

**Affiliations:** 1Group of Inflammatory Diseases, Institut de Recerca Sant Pau (IR Sant Pau), 08041 Barcelona, Spain; mortiz@santpau.cat (M.A.O.); mmulet@santpau.cat (M.M.); rosuna@santpau.cat (R.O.-G.); aguinart@santpau.cat (A.G.-C.); svidal@santpau.cat (S.V.); 2Dermatology Unit, Hospital Quirónsalud, 50012 Zaragoza, Spain; elena.delprado@quironsalud.es; 3Department of Dermatology, Hospital de la Santa Creu i Sant Pau, 08025 Barcelona, Spain; evilarrasa@santpau.cat (E.V.); alopezfe@santpau.cat (A.L.-F.); lpuig@santpau.cat (L.P.); 4Dermatology Unit, Hospital Royo Villanova, 50015 Zaragoza, Spain; drgarcialatasa@yahoo.es; 5Centro Nacional de Analisis Genomico (CNAG), 08028 Barcelona, Spain; marta.gut@cnag.eu (M.G.); anna.esteve@cnag.eu (A.E.-C.); 6Universitat de Barcelona (UB), 08036 Barcelona, Spain; 7Centro de Investigación Biomédica en Red de Enfermedades Hepáticas y Digestivas, Instituto de Salud Carlos III, 28029 Madrid, Spain; 8Department of Cellular Biology, Fisiologia i Immunologia, Universitat Autònoma de Barcelona, 08193 Bellaterra, Spain

**Keywords:** psoriasis, transcriptomic, inflammation, hubs, biomarkers

## Abstract

Psoriasis is a chronic inflammatory skin disease driven by genetic, environmental, and immune factors. While biologics like adalimumab (anti-TNFα) and risankizumab (anti-IL-23) have improved outcomes, patient response variability remains unclear. This study examined immune-related transcriptomic differences between lesional (L) and non-lesional (NL) psoriatic skin, focusing on immune-related hub genes, their plasma levels, and their correlations with severity and treatment response. Patients with moderate-to-severe psoriasis were enrolled before treatment with anti-TNFα (*n* = 16) or anti-IL-23 (*n* = 18). Plasma and paired L and NL skin biopsies were collected for RNA sequencing. Gene ontology enrichment analysis found four immune-related terms enriched in L skin: T-helper 17, granulocyte and lymphocyte chemotaxis, and antimicrobial humoral response. A protein–protein interaction network identified ten immune-related hub genes upregulated in L skin that correlated with clinical severity. Patients with prior treatments expressed distinctive gene profiles. Plasma levels of CCL20 strongly correlated with disease severity. Decision tree models identified CCL20 expression in skin and plasma levels of IL-6 and CXCL8 as candidate predictors for anti-TNFα response. Similarly, skin expression of CXCL8, IL-6, and CXCL10, alongside plasma levels of CCL20, IL-6, and CXCL8, may predict anti-IL-23 response. Ten immune-related hubs may serve as possible biomarkers for disease severity and therapeutic response in psoriasis.

## 1. Introduction

Psoriasis is a chronic immune-mediated skin disease characterized by an abnormal proliferation of keratinocytes and an exaggerated inflammatory response that is prevalent in 2–4% of the global population. It is believed that a combination of genetic, environmental, and immunological factors contributes to its development [1]. At the cellular level, psoriasis is characterized by the accelerated turnover of skin cells, resulting in thickened red scaly plaques. This process is driven by activated T cells and the release of cytokines such as tumor necrosis factor-alpha (TNFα), interleukins (IL-17, IL-23), and interferon-gamma (IFNγ), which promote inflammation and stimulate keratinocyte proliferation [2]. The development of biological agents targeting these cytokines has significantly improved the therapeutic effect and safety of psoriasis treatment. Adalimumab, a monoclonal antibody targeting TNFα, inhibits its interaction with cell surface receptors, thereby suppressing downstream inflammatory pathways and reducing keratinocyte proliferation [3]. Skin RNA sequencing (RNA-seq) analysis showed that adalimumab treatment modulates critical genes involved in epidermal growth and differentiation [4]. Risankizumab, a humanized immunoglobulin G1 monoclonal antibody targeting the p19 subunit of IL-23, inhibits IL-23-dependent cell signaling [5]. Treatment with risankizumab in patients with moderate to severe plaque reduces dermal immune cell infiltration, thereby decreasing tissue inflammation. Gene expression analysis following risankizumab treatment shows a significant reduction in the expression of L skin genes related to the IL-23/IL-17 signaling pathways, suggesting specific targeting of these inflammatory pathways. Notably, the gene expression profile of psoriatic lesions after risankizumab treatment closely resembles that of NL skin, indicating a normalization of gene expression patterns [6].

Biological treatments like anti-TNFα and anti-IL-23 antibodies have revolutionized psoriasis management by targeting key cytokines. However, response rates vary between patients, and the factors influencing these variations are not fully understood and cannot be solely attributed to psoriasis susceptibility genes [5]. Evaluating drug responses before treatment initiation could help not only to prevent adverse side effects but would also contribute to personalized medicine, ensuring that patients receive the most appropriate therapy tailored to their specific immune profile, thereby improving clinical efficacy, patient quality of life, and reducing the long-term costs associated with managing complications and ineffective therapies. Previous studies based on gene expression analysis have identified biomarkers predicting treatment response. For instance, fast responders to guselkumab showed decreased expression of IL-1β, S100A12, MMP9, and CXCL8 genes after treatment initiation [7], while changes involving IFN regulators were noted in response to anti-TNFα therapy [8]. Based on these results, understanding immune-related gene expression profiles seems to be essential for elucidating treatment response in psoriasis and assessing how previous treatments influence these profiles, ultimately impacting clinical outcomes and therapeutic efficacy. The current study aimed to address this hypothesis by comprehensively characterizing transcriptomic differences between L and NL psoriatic skin using RNA-seq. Specifically, the focus was on immune-related differentially expressed genes (DEGs) and key immune-related hub proteins critical for network stability and functionality. Additionally, we determined the plasma levels of immune-related hub proteins and evaluated their associations with disease severity and treatment response. By integrating transcriptomic and plasma biomarker data, the study aimed to provide insights into how previous treatments condition immune profiles and how these profiles can predict treatment outcomes.

## 2. Results

### 2.1. Patient Demographics

We enrolled 34 psoriatic patients prior to the initiation of anti-IL-23 (*n* = 18) or anti-TNFα (*n* = 16) treatment (baseline). Data on demographic variables and previous treatments were collected (Table 1). No differences were observed regarding sex, age, PASI, and BSA at baseline. However, patients who received anti-IL-23 were older at the time of diagnosis and also showed higher PGA than patients treated with anti-TNFα. The main previous treatment for anti-IL-23 patients was biological (66.66%), and for anti-TNFα patients it was topical (50%). Other treatments included apremilast, methotrexate, cyclosporine, and fumarate.

### 2.2. Identification of DEGs in L vs. NL Skin

In the present study, we compared skin RNA expression profiles between paired NL and L skin biopsies from psoriatic patients and detected 18,729 protein-coding mRNAs. A volcano plot was used to show the overall distribution of all DEGs when comparing L vs. NL skin (Figure 1A).

A paired comparison of gene expression between L and NL skin biopsies yielded 1492 DEGs, including 738 L-increased DEGs (log_2_FC ≥ 0.58 and *p* adjusted < 0.05) and 754 L-decreased DEGs (log_2_FC ≤ −0.58 and *p* adjusted < 0.05). To evaluate the biological characteristics and enrichment pathways of the identified DEGs, GO enrichment and KEGG enrichment analysis were conducted using ShinyGO 0.80. In Appendix A, the upregulated and downregulated DEGs were analyzed separately, selected by FDR, and sorted by fold enrichment. Principal component analysis (PCA) revealed clear segregation of NL from L skin (Figure 1B). The median expression of all DEGs (represented as score) showed a significantly increased score in L skin compared to NL skin (*p* < 0.001) (Figure 1C). A heatmap displaying upregulated and downregulated genes is presented in Figure 1D.

Gene Ontology enrichment analysis of immune-related processes was conducted with the 1492 DEGs using ClueGO v2.5.10 in Cytoscape v3.10.1 with “GOImmuneSystemProcess-EBI-UniProt-GOA-ACAP-ARAP-25.05.2022” as the input ontology file (Figure 1E). Using the 114 identified immune-related genes, the network revealed distinct immune-related clusters, including T-helper 17 type immune response (53.33%), granulocyte chemotaxis (26.67%), antimicrobial humoral response (13.33%), and lymphocyte chemotaxis (6.67%), illustrating a coordinated immune activation.

### 2.3. Identification of Skin Immune Hub Genes

The selected 114 immune-related genes were imported into the STRING database to construct the protein–protein interaction network. The top 10 hub genes with a high degree of connectivity were selected using the cytoHubba plugin v0.1 (MCC algorithm) (Figure 2A). These highly connected genes interacted extensively within the network, serving as central regulators of key biological processes. These included cytokines (IL-6, IFNγ, and IL-1β), chemokines (CXCL10, CXCL9, CXCL13, CXCL8, CCL3, and CCL20), and a receptor for chemokines (CCR7). A total of seven immune-related hubs were consistently identified across other four additional algorithms (Degree, EPC, MNC, and Closeness).

Figure 2B shows the upregulation of the top 10 hub genes in L skin compared to NL skin. There were strong positive correlations among the expression levels of the 10 hub genes in L skin from psoriatic patients (Figure 2C). By contrast, in NL skin, only IL-1β showed positive correlations with CCR7 (r = 0.548, *p* < 0.001) and CCL20 (r = 0.487, *p* = 0.003), while CXCL10 positively correlated with CXCL9 (r = 0.882, *p* < 0.001). IL-6 was the only gene whose expression levels were significantly correlated between L and NL skin (r = 0.450, *p* = 0.007). At baseline, patients with severe PASI (>10) showed a trend toward higher expression of CXCL10 (*p* = 0.060) and CXCL9 (*p* = 0.060) in NL skin compared to those with mild PASI (≤10) (Figure 2D). Elevated CXCL8 (*p* = 0.046) and IL-1β (*p* = 0.046) expression in NL skin and increased IL-6 expression in L skin (*p* = 0.041) were associated with greater BSA severity (Figure 2E). No differences related to PGA at baseline were found.

### 2.4. Previous Treatment Conditioned Differential Skin DEGs and Skin Immune Hubs Expression

Table 2 summarizes the demographic characteristics of psoriatic patients based on previous treatment (topical, biological, and other). No significant differences were observed among the three groups regarding PASI, BSA, and PGA at baseline.

In NL skin from patients previously treated with topical therapy, 54 genes were upregulated and 112 genes were downregulated compared to NL skin from patients previously treated with biological therapy (Figure 3A). An analysis of GO showed that the DEGs were mainly related to oxygen transport and G-protein coupled receptor (GPCR) ligand binding. By contrast, in L skin, 47 genes were upregulated and 253 genes were downregulated in patients previously treated with topical therapy compared to those treated with biological therapy (Figure 3B).

An analysis of GO showed that the DEGs were mainly related to the regulation of membrane potential and neuroactive ligand–receptor interactions. The expression of hub genes categorized by previous treatment (Figure 3C) revealed that, in NL skin, only CXCL8 showed significant differences between topical, biological, and other treatments (*p* = 0.019). In L skin, statistical trends were observed for differences in CXCL8 (*p* = 0.090) and IL-6 (*p* = 0.090) expression (Figure 3D).

### 2.5. Response to Anti-TNFα

Patients were categorized based on their response after six months of anti-TNFα treatment (Table 3). Responders (R) were defined as patients achieving PASI = 0 (*n* = 7), whereas non-responders (NonR) did not achieve PASI = 0 (*n* = 9) (Figure 4A). Although this is a stringent criterion, we adopted it to define an ideal response, supported by real-world evidence linking complete clearance to better long-term outcomes [9]. No significant differences were found in sex, age at baseline, age at diagnosis, PASI, BSA, PGA, or PASI/BSA ratio between R and NonR patients. Among R patients, 71% (5/7) received topical treatment, compared to 33% (3/9) of NonR patients. Conversely, 29% of R patients received other previous treatments, compared to 67% of NonR patients (Figure 4A).

We found significant differences in the expression of immune-related hub genes between NL and L skin in both R and NonR patients (Figure 4B). However, no differences in the expression levels of hub genes were found in NL or L skin between R and NonR patients, even when considering only patients who had received previous topical therapy. A decision tree model based on the expression of the 10 hub genes was applied to distinguish between R and NonR patients who had previously received topical treatment. R patients were identified in 80% of cases (*n* = 4) based on CCL20 expression in L skin (log_2_cpm > 1.94), with a classification purity of 100% (Figure 4C). The confusion matrix is shown in Appendix A.

### 2.6. Response to Anti-IL-23

Patients were categorized based on anti-IL-23 response after three and six months of treatment (Table 4). One patient previously treated with anti-p19 was excluded from this analysis. Super-responders (SR) were defined as patients achieving PASI = 0 at both three and six months (*n* = 8), whereas NonR patients did not achieve PASI = 0 at three or six months (*n* = 9) post-initiation of anti-IL-23 treatment (Figure 5A). Several studies conducted in real clinical practices have proposed this definition of SR, underlining its growing relevance in therapeutic decision making [10].

No significant differences were observed in sex, age at diagnosis, age at baseline, PASI, BSA, PGA, or PASI/BSA ratio at baseline between SR and NonR patients. Among SR patients, 50% (4/8) had previously received biological treatment (Figure 5A), compared to 78% of NonR patients. Conversely, 50% of SR patients had received other previous treatments, compared to 22% of NonR patients. We found significant differences in the expression of immune-related hub genes between NL and L skin in both SR and NonR patients, except for CXCL10 and IL-6 (Figure 5B). There were no significant differences in the expression of immune-related hub genes between SR and NonR patients in both L and NL skin. However, when comparing only within the group of patients who had previously received biological treatment, the expression of IL-6 in L skin was lower in SR compared to NonR patients (*p* = 0.012). A decision tree model based on the expression of the 10 identified hub genes was applied to distinguish between SR and NonR patients who had previously received biological treatment. SR patients were identified in 75% of cases (*n* = 3) based on CXCL8 expression in L skin (log_2_cpm < 4.00), IL-6 expression in NL (log_2_cpm ≤ 2.97), and CXCL10 expression in L skin (log_2_cpm > 1.58), with a classification purity of 100% (Figure 5C). The confusion matrix is shown in Appendix A. In the anti-IL-23 group, the prior biologic exposure was heterogeneous, but including the type of biologic as a covariate in the model did not improve classification performance.

### 2.7. Plasma Immune Hubs and Their Relationships with Clinical Psoriasis Characteristics

We analyzed the levels of CXCL10, CXCL8, IL-6, CCL20, and CXCL13 in the plasma of psoriatic patients and HD (Figure 6A). We only included a selection of the immune hubs because the published data indicate that CXCL10 is a product of IFN-γ, while IL-6, CXCL13, CCL20, and CXCL8 are products of IL-1β. Patients exhibited higher plasma levels of CCL20 (*p* = 0.015), along with trends toward higher levels of IL-6 (*p* = 0.087) and CXCL13 (*p* = 0.050) compared to HD. No significant differences were observed for CXCL10 and CXCL8.

In Figure 6B, we observed a positive correlation between plasma levels of CXCL8 and IL-6 (r = 0.676, *p* < 0.001) in patients. We found no significant correlations between plasma levels and gene expression in NL skin (Figure 6C). However, several correlations were identified between plasma levels and L skin (Figure 6D and Appendix A): plasma CXCL8 levels correlated negatively with IL-6 expression (r = −0.374, *p* = 0.029); plasma IL-6 levels correlated negatively with CXCL9 (r = −0.457, *p* = 0.008) and CXCL13 (r = −0.413, *p* = 0.018) expression; plasma CXCL13 levels correlated negatively with CXCL10 (r = −0.429, *p* = 0.011) and CXCL9 (r = −0.398, *p* = 0.019) expression; and plasma CCL20 showed a modest but statistically significant correlation with CCL20 expression (r = 0.351, *p* = 0.045). When stratifying the plasma levels of immune-related hubs by previous treatments (topical, biologic, and other), a statistical trend was observed for IL-6 (*p* = 0.069) (Figure 6E). Regarding baseline clinical characteristics, elevated plasma levels of CCL20 were associated with increased BSA severity (severe: 62.68 ± 61.84 pg/mL; mild: 38.83 ± 57.13 pg/mL; *p* = 0.010) and PGA (PGA > 3: 66.06 ± 69.19 pg/mL; PGA < 3: 42.42 ± 45.63 pg/mL; *p* = 0.047).

### 2.8. Plasma Immune Hub Expression Conditions Anti-TNFα and Anti-IL-23 Response

No significant differences in the plasma levels of immune-related hub proteins were observed between R and NonR patients treated with anti-TNFα (Figure 7A), or between SR and NonR patients treated with anti-IL-23 (Figure 7B), either when considering all patients or when specifically analyzing those previously treated with topical (Figure 7A) or biologic therapies (Figure 7B).

We applied a decision tree model based on the plasma levels of five hubs to discern between R and NonR patients to anti-TNFα who had previously received topical treatment (Figure 7C). The model revealed that R patients were identified in 60% of cases (*n* = 3) based on IL-6 plasma levels > 27.75 pg/mL. The remaining 40% were classified based on CXCL8 levels ≤ 6.99 pg/mL (purity 100%) (*n* = 2). Figure 7D shows a decision tree model applied to patients with prior biologic therapy to distinguish SR from NonR in response to anti-IL-23, based on the plasma levels of five hub proteins. The model classified 50% of SR patients (*n* = 2) as having CCL20 levels < 82.94 pg/mL and IL-6 levels ≤ 0.20 pg/mL. The remaining 50% (*n* = 2) were further classified based on CXCL8 plasma levels > 8.36 pg/mL, with a classification purity of 66.7%. The confusion matrices are shown in Appendix A.

## 3. Discussion

In this exploratory study, we used RNA-seq to compare the immune-related gene expression profiles of L and NL skin from 34 psoriatic patients. Although skin biopsies are invasive and not ideal for routine clinical use, in our study they served primarily to uncover mechanistic insights and identify candidate plasmatic biomarkers. Our aim was to identify immune-related hub genes with high connectivity that may play a central role in the immune network. Our findings suggest that previous treatments significantly shape immune gene expression in psoriatic skin and that these profiles may predict anti-TNFα and anti-IL-23 responses. Notably, immune hubs such as CXCL8, CXCL10, CCL20, and IL-6 emerged as potential determinants of therapeutic response. Among them, CCL20 showed moderate concordance between L skin and plasma levels, predictive value for response to anti-TNFα therapy, and elevated concentrations in psoriatic patients compared to healthy donors.

Our approach specifically targeted immune-related hubs, as the DEGs showed significant enrichment in immune-related pathways [11]. We identified 10 key immune hub genes—notably upregulated in L skin—that may play central roles in sustaining inflammation and driving psoriasis pathogenesis. Our results align with other studies using comparable strategies. For example, bioinformatic analyses comparing L skin with healthy donors have identified five of the same hub genes (CXCL8, CXCL10, CCL20, CXCL9, and CCR7) [12], supporting their biological relevance. One study has also identified upstream transcription factors related to enzymatic activity in L skin that may contribute to the expression of IL-6, IL-1β, CXCL8, and CXCL10 [13], and another study has found 17 hubs involved in the cell cycle and IFNα-induced genes when comparing L with NL skin in mild and moderate psoriasis [14]. More recently, lysosomal hubs have been proposed as potential diagnostic markers [15]. Together, these findings highlight the complexity of the gene network involved in psoriasis, implicating multiple biological pathways in disease progression and therapeutic targeting.

We found that higher expression of CXCL10, CXCL9, CXCL8, and IL-1β in NL skin correlated with greater disease severity. This suggests that NL skin—while a practical internal control—differs molecularly from healthy skin, harboring early inflammatory mechanisms. CXCL8 and CXCL13 have already been implicated in early disease stages, with CXCL8 upregulated in mild psoriasis [16] and CXCL13 expressed by lesion-infiltrating T cells, correlating with IL-17A levels [17]. Furthermore, CXCL9 and CXCL10 are also upregulated in NL skin, potentially contributing to the formation of new lesions [18]. Altogether, these data support the notion that NL skin represents a “pre-psoriatic” state, characterized by altered innate immunity, increased epidermal differentiation, and reduced expression of barrier proteins, contributing to disease progression [19,20,21,22]. Further studies should evaluate whether NL profiling could identify patients at risk of developing more severe disease.

We also observed that prior treatments influenced expression in both L and NL skin. Patients treated with topical agents exhibited different gene expression signatures compared to those receiving biologics. In L skin, topical treatments appear to affect neuronal signaling and skin barrier function while, in NL skin, they may alter skin metabolism and intercellular communication pathways—even in areas without visible disease. Interestingly, prior treatment also altered the expression of immune-related hub genes, such as CXCL8 and IL-6. Several studies have reported changes in psoriatic skin induced by various treatments. For instance, therapies targeting IL-23 and IL-12/IL-23 have been associated with changes in keratin expression and barrier integrity in L skin four weeks after treatment initiation [23]. Etanercept treatment upregulated DEGs related to inflammation and immune responses in L skin after 12 weeks, while normalizing gene expression in NL skin [8,24]. Similarly, fumaric acid esters have been shown to regulate key transcription factors that are critical in epidermal development and Th2 and Th17 immune pathways [25]. These findings suggest that therapeutic history shapes the local immune environment, potentially impacting responsiveness to subsequent therapies. Comprehensive longitudinal studies assessing gene expression before and after each therapy could clarify treatment-specific molecular changes and improve personalized treatment strategies.

Extending our analysis to plasma, we observed changes in circulating immune-related hub genes in psoriatic patients compared to healthy donors, bridging local and systemic immune environments [26]. Interestingly, we found inverse correlations between several hub genes (CXCL10, CXCL9, CXCL13, and IL-6) in L skin and plasma. These discrepancies may indicate compartmentalized cytokine production, tissue-specific regulation, or local retention of inflammatory mediators. A modest but statistically significant positive correlation between CCL20 levels in L skin and plasma indicated CCL20 as a potential plasma biomarker for severe psoriasis. CCL20, produced locally in psoriatic lesions, likely recruits CCR6+ Th17 cells [27], and it is spilled into the systemic circulation in severe cases. Consistent with this, plasma CCL20 levels correlated with clinical severity measures, including BSA and PGA, reinforcing its potential as a non-invasive biomarker of disease severity and treatment response. Moreover, the pattern of skin–plasma correlations varied depending on prior treatment, indicating that previous therapies influence both local and systemic immune environments.

Finally, using a decision tree approach [28], we identified different immune hubs that predict treatment response based on prior therapy. Among patients with prior topical treatment, CCL20 expression in L skin distinguished R from NonR to anti-TNFα therapy. This finding aligns with data showing TNFα as a strong inducer of CCL20 [29]. The integration of skin and plasma analysis of responders to anti-TNFα therapy suggested two patient subgroups: (1) those with high plasma IL-6, whose therapy reduced both local (CCL20) and systemic inflammation (IL-6), and (2) those with low plasma levels of IL-6 and CXCL8, but high expression of skin CCL20, suggesting a dominant localized TNFα-driven inflammation. In patients previously treated with biologics, CXCL8 and CXCL10 expression in L skin and IL-6 expression in NL skin were predictive of anti-IL-23 response. These patients may have shifted from a Th17-driven response to a Th1-dominated environment, improving outcomes for patients with higher baseline CXCL10 levels [30,31,32,33]. Supporting this, plasma CCL20 was undetectable in half of the IL-23- SR patients, indicating a reduced role of systemic TNFα.

While our exploratory study provides important mechanistic insights into psoriasis and treatment responses, it has limitations. These include a small sample size, a single time point for RNA-seq analysis, short follow-up, and limited power to apply methods such as random forest modeling. We also lacked data on lifestyle factors, comorbidities, and concomitant medications. Additionally, while NL skin serves as a valuable intra-individual control, it may limit the ability to detect differences with truly healthy skin. Despite these limitations, our findings emphasize the value of immune hub profiling in understanding disease mechanisms and optimizing treatment.

In conclusion, this study identified immune-related hub genes and proteins in the skin and plasma of psoriatic patients that may be relevant for predicting response to biologic therapies. Particularly, CCL20 expression in skin and plasma levels of IL-6 and CXCL8 as candidate predictors for anti-TNFα response, and skin expression of CXCL8, IL-6, and CXCL10, alongside plasma levels of CCL20, IL-6, and CXCL8, may predict anti-IL-23 response.

We propose that identifying predictive biomarkers—while considering prior treatment history—may guide more tailored therapeutic strategies. Larger prospective studies are needed to validate these candidate biomarkers and confirm their clinical utility in treatment decisions.

## 4. Materials and Methods

### 4.1. Patient Cohort and Samples

We recruited 34 patients with conventional plaque psoriasis at two centers (18 from Hospital de Sant Pau, Barcelona, and 16 from Hospital Royo Villanova, Zaragoza). Adults (>18 years) with moderate-to-severe plaque psoriasis (Psoriasis Area and Severity Index, PASI ≥ 10; Body Surface Area, BSA ≥ 10%; Physician Global Assessment, PGA ≥ 3) were included if they were eligible for anti-TNFα or anti-IL-12/IL-23 therapy according to clinical guidelines. This study was an exploratory and hypothesis-generating investigation that included patients who had previously received classical systemic or biological treatment, as well as treatment-naive patients without prior systemic treatment for psoriasis, and no active infections or significant comorbidities contraindicating biologic treatment.

Two 4-mm punch biopsy specimens were obtained from each patient’s skin prior to treatment initiation (baseline) with anti-IL-23 (*n* = 18) or anti-TNFα (*n* = 16). One biopsy was taken from an L area with visible signs of psoriasis, and the other was taken from NL skin without psoriatic signs. Biopsy samples were immediately transferred to RNAlater^®^ solution (Ambion, Thermo Fisher, Austin, TX, USA) and stored at 4 °C for 24 h, followed by storage at −80 °C until RNA extraction. Simultaneously, blood samples from the same patients were collected to obtain plasma, which was stored at −80 °C until use. Plasma samples from 18 age- and sex-matched healthy donors (HD) were obtained for comparison. The study was conducted in accordance with the Declaration of Helsinki, and signed informed consent was obtained from each patient. All experimental protocols were approved by a named institutional and/or licensing committee.

### 4.2. RNA Sequencing

Biopsy specimens were homogenized using TissueRuptor (Qiagen, Hilden, Germany), and total RNA was extracted using the Qiagen RNeasy^®^ Fibrous Tissue Mini Kit (Qiagen), following the manufacturer’s protocol. RNA integrity was assessed using the RNA 6000 Nano Bioanalyzer 2100 assay (Agilent Technologies Inc., Santa Cruz, CA, USA), with samples having an average RNA integrity number (RIN) of ≥9. RNA quantity was measured using the Qubit^®^ RNA BR Assay kit (Thermo Fisher Scientific, Waltham, MA, USA).

The RNA-seq libraries were prepared using the KAPA Stranded mRNA-Seq Kit for the Illumina^®^ Platform (Roche, Basel, Switzerland), starting with 500 ng of total RNA as input. Library quality was assessed using the Agilent 2100 Bioanalyzer with the DNA 7500 assay (Agilent Technologies Inc., Santa Cruz, CA, USA). Samples were randomized across batches to minimize batch effects. Sequencing was performed on a NovaSeq 6000 (Illumina, San Diego, CA, USA) with 2 × 51 bp reads and dual indexing. On average, 40–50 million paired-end reads were generated per sample. When initial sequencing depth was insufficient, technical replicates were sequenced to meet the required depth and subsequently aggregated at the counts level. Image analysis, base calling, and quality scoring were conducted using RTA 3.4.4 software. RNA-seq reads were aligned to the human reference genome GRCh38 using STAR/2.7.8a with ENCODE parameters. Gene quantification was performed using RSEM/1.3.0 with the GENCODE v40 annotation. Post-sequencing PCA and hierarchical clustering revealed no significant batch effects, and all samples passed QC based on mapping rates and read quality. We applied TMM normalization (Trimmed Mean of M-values) using the edgeR package v3.38.4 (accessed on 19 May 2022) prior to voom transformation of the raw counts, as recommended for use with the limma-voom pipeline. The voom-transformed data were then used for linear modeling and empirical Bayes moderation of standard errors in limma. Differential expression analysis was performed using the limma R package v3.52.4 (accessed on 19 May 2022) with voom transformation, accounting for repeated measures with the duplicate. Multidimensional scaling plots were generated using plotMDS from limma. Significant differentially expressed protein-coding genes (DEGs) between conditions L vs. NL were identified based on an adjusted *p*-value < 0.05 and log_2_ fold change (FC) ≥ 0.58 or ≤−0.58. The adjusted *p*-values reported in the results were calculated using the false discovery rate (FDR) correction according to the Benjamini–Hochberg procedure. DEGs between topical vs. biologic pretreatments were identified with a *p*-value < 0.05 and log_2_FC ≥ 1 or ≤−1. Heatmaps were constructed using http://www.heatmapper.ca/ (Edmonton, Canada) (accessed on 16 June 2024) with average linkage clustering and Spearman’s rank correlation [34].

### 4.3. Functional Enrichment Analysis

The functional enrichment analysis of DEGs (Gene Ontology and Kyoto Encyclopedia of Genes and Genomes (KEGG)) was performed using ShinyGO 0.80 (http://bioinformatics.sdstate.edu/go) (Brookings, SD, USA) (accessed on 21 May 2024) with an FDR cutoff of 0.05, showing up to 20 pathways and considering pathway sizes between 2 and 5000 genes [35]. To determine the DEGs related to the immune system, a gene ontology (GO) enrichment analysis of immune-related processes was conducted using ClueGO v2.5.10 in Cytoscape v3.10.1 (Seattle, WA, USA) (accessed on 25 March 2024), with “GOImmuneSystemProcess-EBI-UniProt-GOA-ACAP-ARAP-25.05.2022” as the input ontology file [36]. A minimum of three genes per cluster and significant enrichment terms (*p* < 0.05) were visualized. Each node represents a GO term from the immune system process, with node size indicating GO term significance (smaller *p*-value corresponds to a larger node size). Edges (lines) between nodes indicate the presence of common genes, where a thicker line implies a larger overlap. GO terms are classified into several functional groups.

### 4.4. Protein–Protein Interaction (PPI) Network Construction and Immune-Related Hub Genes

The list of immune-related genes from “GOImmuneSystemProcess-EBI-UniProt-GOA-ACAP-ARAP-25.05.2022” was used to predict the protein–protein interaction (PPI) network using the Search Tool for the Retrieval of Interacting Genes/Proteins (STRING) (Lausanne, Switzerland) (accessed on 29 on January 2024), an online database of known and predicted protein interactions. Interactions with a combined score > 0.4 were considered statistically significant. The use of a STRING score > 0.4 is considered a medium confidence threshold that balances sensitivity and specificity [37]. Cytoscape was used to visualize molecular interaction networks, and cytoHubba, a plugin of Cytoscape, was used to define the top 10 immune-related hubs. CytoHubba assigns a value to each gene through a topological network algorithm, sequentially identifies hub genes, and ranks nodes according to their properties, providing a simple interface to analyze networks with scoring methods in a preloaded PPI network. To identify the 10 most critical nodes, immune-related nodes were ranked using the Maximal Clique Centrality (MCC) algorithm in cytoHubba because of its high sensitivity in detecting strongly connected central genes within functional subnetworks [38]. To ensure the consistency of the identified hubs with the MCC algorithm, the results were cross-validated using Degree, EPC, MNC, and Closeness algorithms.

### 4.5. ELISAs

Plasma levels of CXCL10 (R&D Systems, Minneapolis, MN, USA), CXCL8, IL-6 (Mabtech, Nacka Strand, Sweden), CCL20 (BioLegend, San Diego, CA, USA), and CXCL13 (Peprotech, Rocky Hill, NJ, USA) were quantified using standard curves provided by their respective ELISA kits, following the manufacturer’s instructions.

### 4.6. Statistical Analysis

Percentages and continuous variables were described using mean ± SD or median plus interquartile range (IQR), according to data distribution (normality). The Kolmogorov–Smirnov test was applied to test for normal data distribution. The Student’s *t*-test, the unpaired *t*-test with Welch’s correction, and the Mann–Whitney U test were used to compare two groups, depending on the distribution of the data (parametric or non-parametric, respectively). Paired *t*-tests and Wilcoxon matched pairs tests were used to compare NL and L skin from the same patient for parametric and non-parametric data, respectively. For multiple group comparisons, one-way ANOVA followed by Tukey’s post hoc test was used for parametric data, and the Kruskal–Wallis test followed by Dunn’s post hoc correction was applied for non-parametric data. *p*-values were adjusted (FDR correction) using the Benjamini–Hochberg method in multiple group comparisons. Spearman correlation coefficients and associated *p*-values were calculated using the rcorr() function from the Hmisc package (v 5.2-3) in R (v 4.3.1). Only the upper triangular portion of the correlation matrix was retained for visualization. The matrix was plotted using the ggplot2 package (v 3.5.2) in R (v 4.3.1), with color gradients representing correlation strength and asterisks indicating statistical significance (*p* ≤ 0.05, 0.01, 0.001) [39]. A classification decision tree model was constructed to determine the best clustering variables according to the C&RT method (maximum tree depth = 3). Due to the heterogeneity in prior treatments and their potential impact on baseline immune status, the decision tree analyses were stratified by treatment group: patients in the anti-TNFα group had only received topical therapies, while those in the anti-IL-23 group had a history of biologic treatments. Statistical analyses were performed using GraphPad Prism Version 7 (GraphPad Software, San Diego, CA, USA) and XLSTAT 2021 software (Addinsoft, Paris, France).

## Figures and Tables

**Figure 1 ijms-26-08118-f001:**
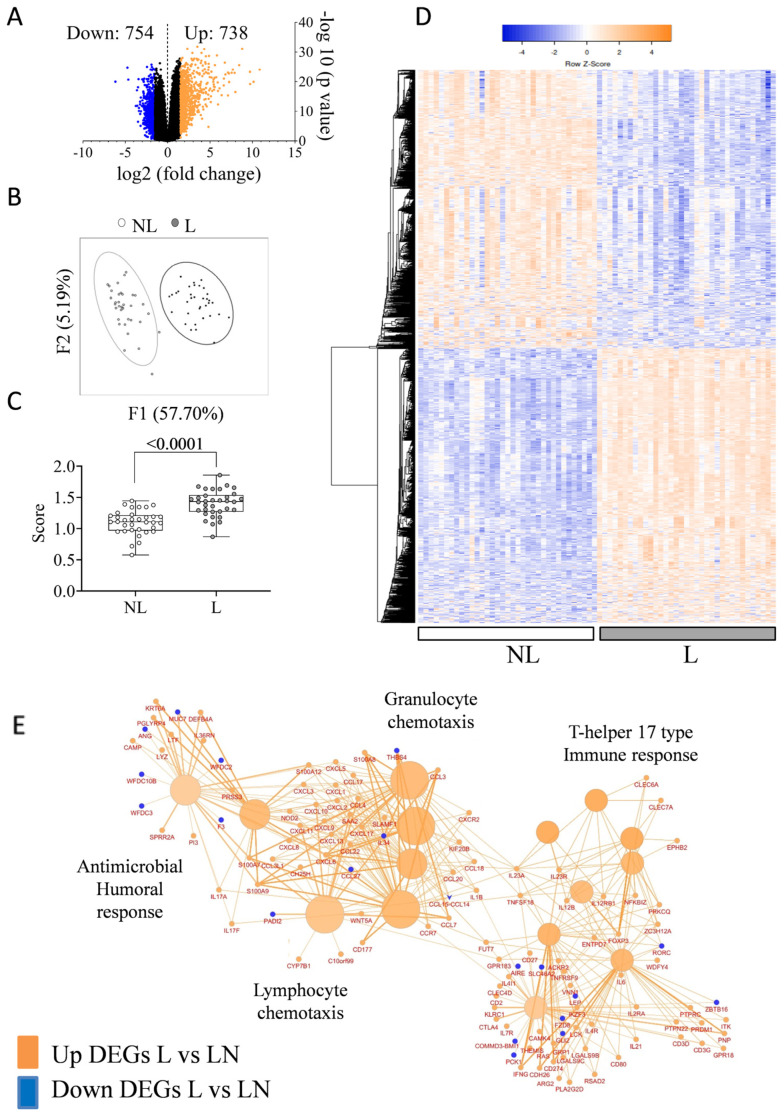
Transcriptomic changes in L vs. NL psoriatic skin. (**A**) Volcano plots showing log_2_FC vs. adjusted *p*-value for protein-coding transcripts. Blue: downregulated genes (log_2_FC ≤ −0.58), orange: upregulated genes (log_2_FC > 0.58); *p* adjusted < 0.05. Non-significant genes in black; (**B**) Principal component analysis (PCA) distinguishing NL (white) from L (gray) skin; (**C**) DEG-based score index calculated from median expression of all DEGs; (**D**) Heatmap of 1492 DEGs in L vs. NL; (**E**) Immune-related GO enrichment of DEGs using Cytoscape v3.10.1. Terms grouped by Kappa statistics (3 final groups; *p* < 0.05, Bonferroni correction). Node size reflects significance. Large circles = GO terms; small = genes. Colors indicate up- or downregulated gene enrichment; edge thickness reflects gene overlap; gradient shows gene proportion per term.

**Figure 2 ijms-26-08118-f002:**
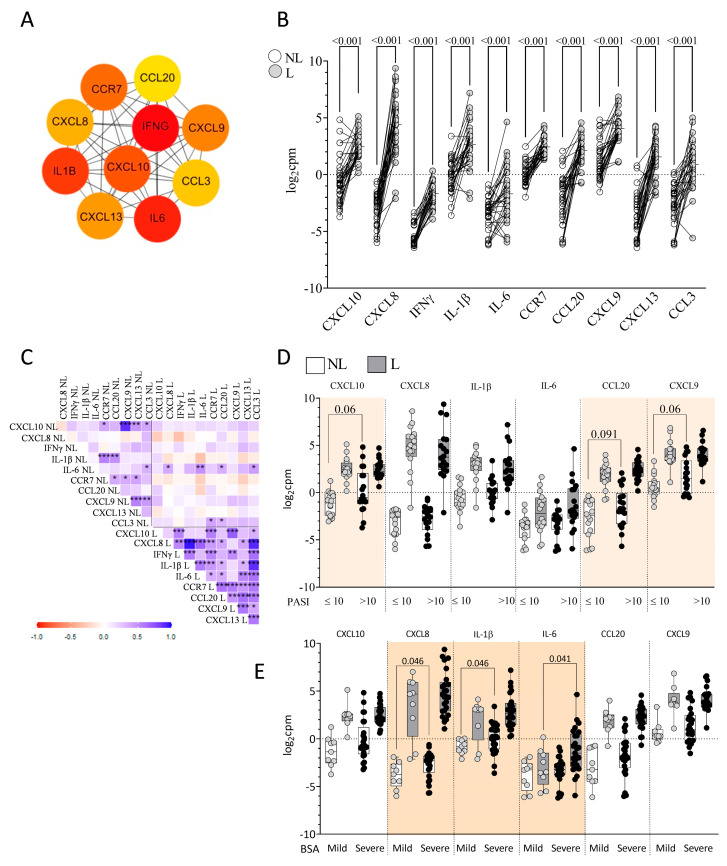
Immune-related hub genes in psoriatic skin. (**A**) Top 10 immune-related hub genes identified using the MCC algorithm in cytoHubba (from 114 immune-related DEGs). Colors represent connectivity (red > orange > yellow); (**B**) Log_2_cpm expression of hub genes in NL (white, *n* = 34) and L (gray, *n* = 34) skin; (**C**) Correlation matrix of hub gene expression in NL and L skin (* *p* < 0.05; ** *p* < 0.01; and *** *p* < 0.001); (**D**) Expression of hub genes by PASI and (**E**) by BSA severity. Paired *t*-tests used in (**B**); unpaired *t*-tests or Welch’s correction in (**D**,**E**).

**Figure 3 ijms-26-08118-f003:**
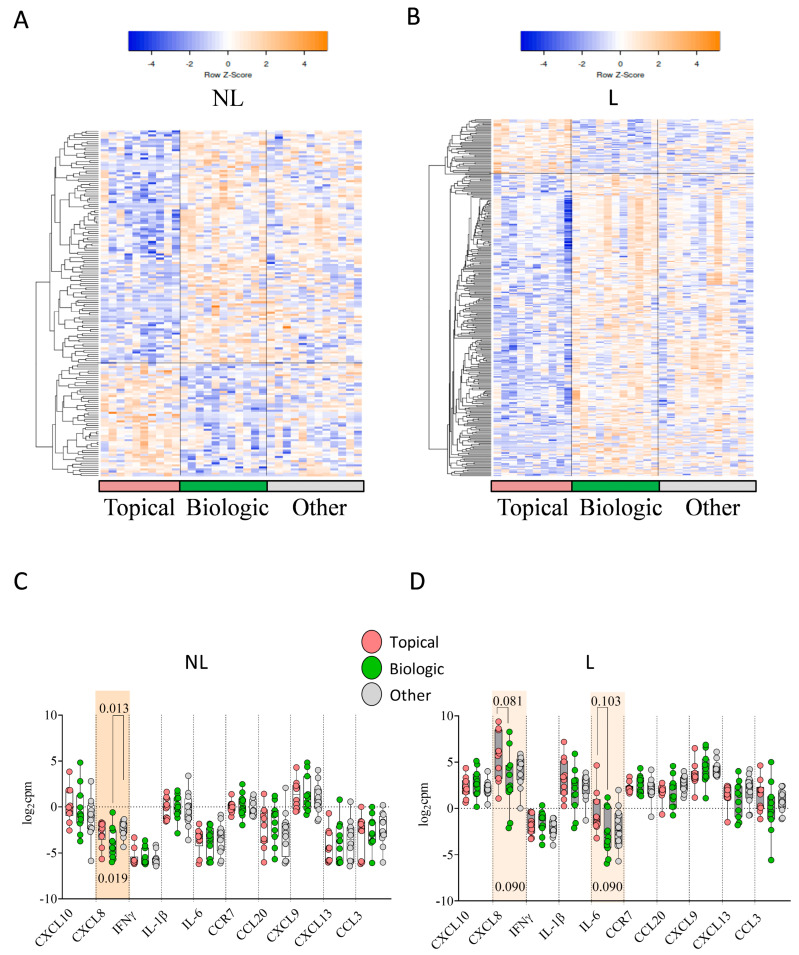
Transcriptomic differences in NL and L psoriatic skin by previous treatment. (**A**) Heatmap of 166 DEGs in NL skin and (**B**) heatmap of 300 DEGs in L skin across treatment groups: topical (pink, *n* = 10), biologic (green, *n* = 11), other (gray, *n* = 12); (orange = upregulated and blue = downregulated); Expression of hub genes in NL (**C**) and L (**D**) skin by previous treatment. Kruskal–Wallis test with Bonferroni correction was used.

**Figure 4 ijms-26-08118-f004:**
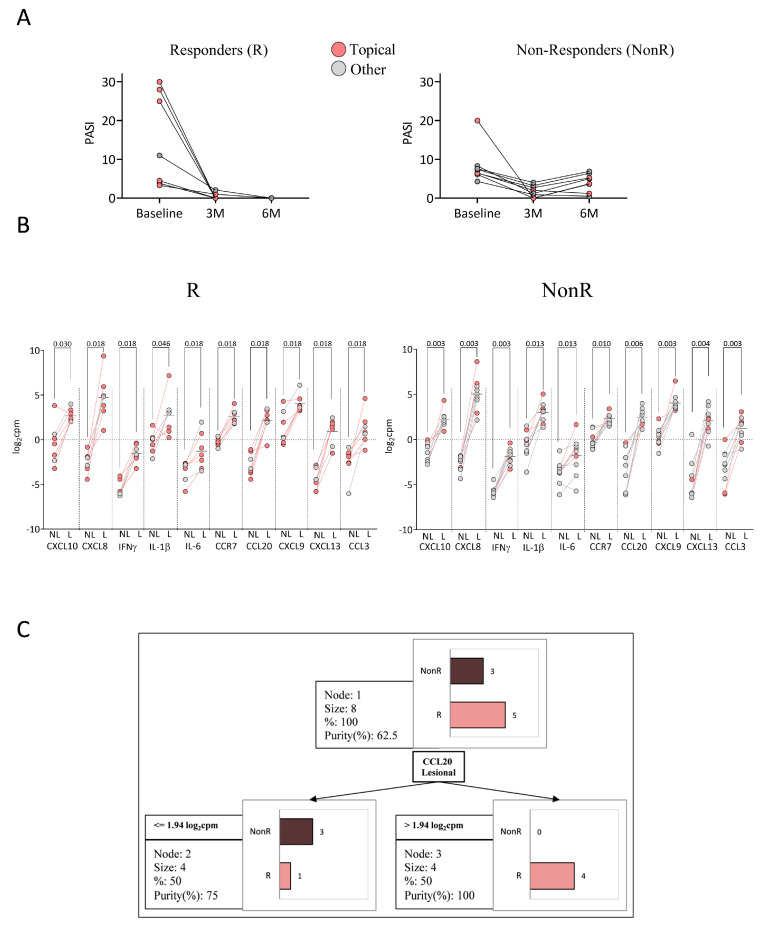
Anti-TNFα treatment response classification. (**A**) Responders (R, *n* = 7) vs. non-responders (NonR, *n* = 10) colored by previous treatment; (**B**) Hub gene expression in NL and L skin of R and NonR patients by prior treatment (Wilcoxon matched-pairs tests); (**C**) Classification tree based on 10 hub genes from patients with prior topical treatment. C&RT method identified CCL20 expression in L skin as main classifier (87.5% accuracy, 80% sensitivity, and 100% specificity).

**Figure 5 ijms-26-08118-f005:**
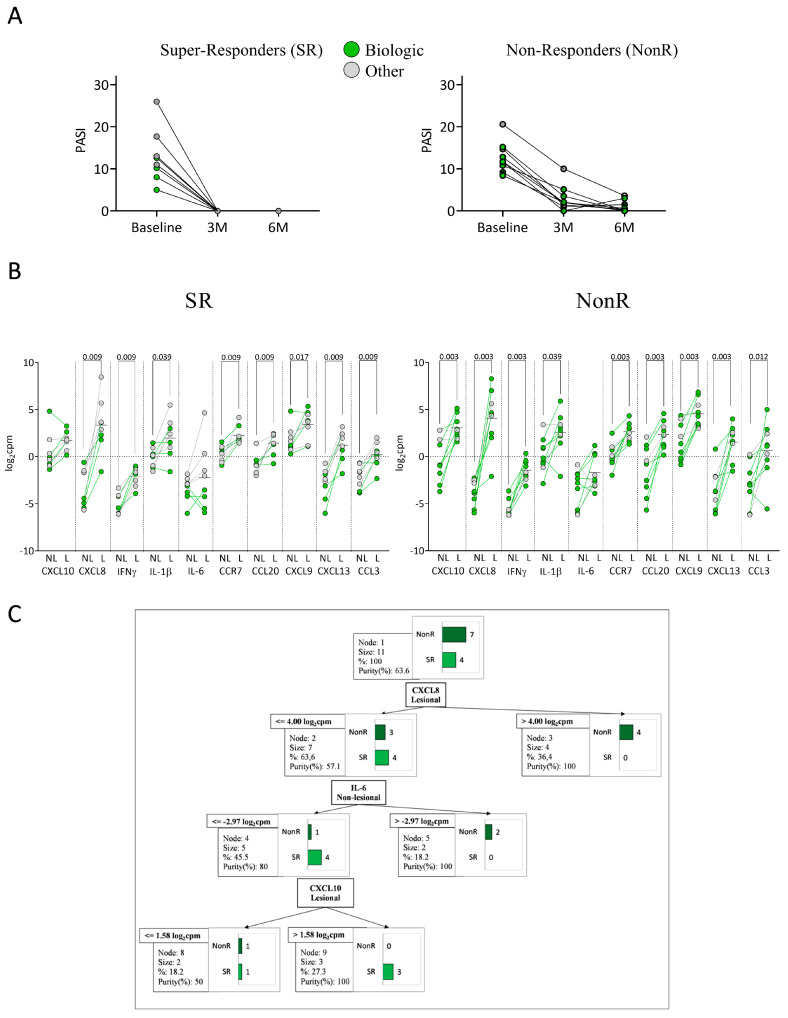
Anti-IL-23 treatment response classification. (**A**) Super-responders (SR, *n* = 8) vs. non-responders (NonR, *n* = 9) colored by previous treatment; (**B**) Hub gene expression in NL and L skin by previous treatment (Wilcoxon matched-pairs tests); (**C**) Classification tree based on 10 hub genes from patients with prior biologic treatment. C&RT method identified CXCL8, CXCL10 (L skin), and IL-6 (NL skin) as classifiers (90% accuracy, 75% sensitivity, 100% specificity).

**Figure 6 ijms-26-08118-f006:**
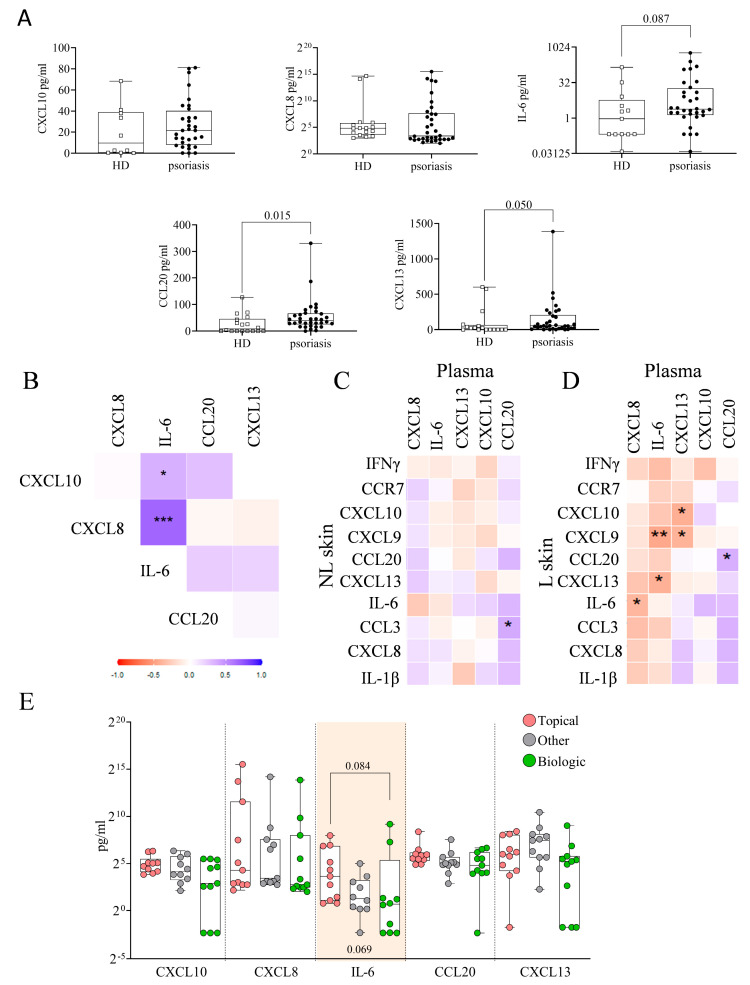
Plasma levels of immune-related hub proteins. (**A**) Hub protein levels in plasma from HD (*n* = 18) and psoriatic patients (*n* = 34) (Mann–Whitney test); (**B**) Correlation matrix of plasma hub proteins (* *p* < 0.05, ** *p* < 0.01, *** *p* < 0.001); Correlation matrices between plasma levels and hub gene expression in NL (**C**) and L (**D**) skin; (**E**) Plasma hub levels by previous treatment (Kruskal–Wallis test with Dunn’s post hoc).

**Figure 7 ijms-26-08118-f007:**
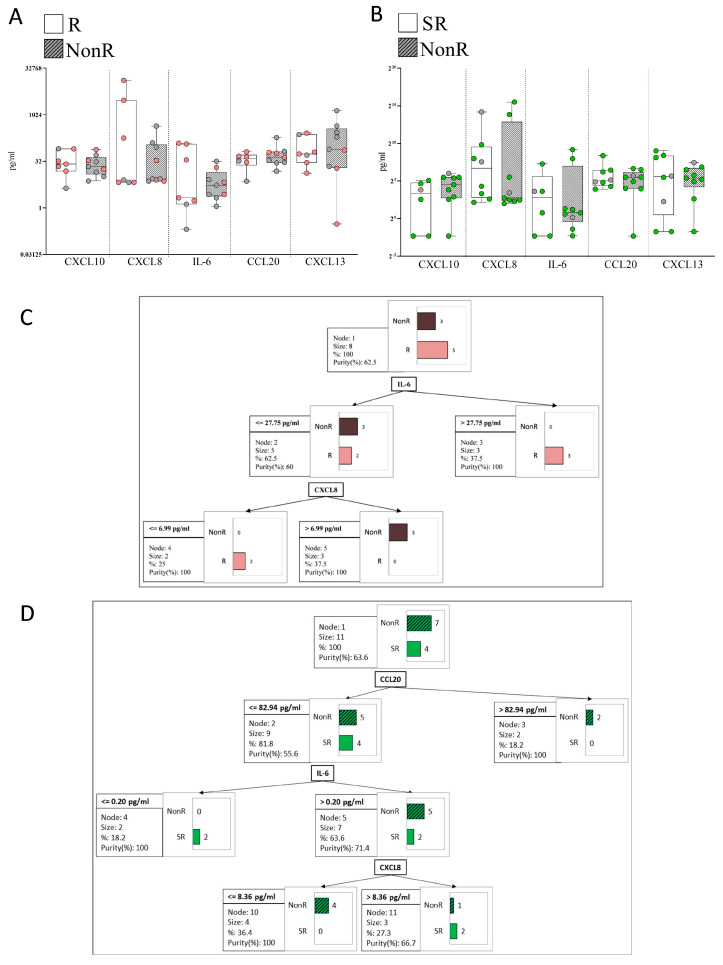
Plasma hub levels and responses to anti-TNFα or anti-IL-23. (**A**) Responders (R, *n* = 7) vs. non-responders (NonR, *n* = 10) to anti-TNFα; (**B**) Super responders (SR, *n* = 8) vs. NonR (*n* = 9) to anti-IL-23; (**C**) Classification tree model for anti-TNFα response based on five hub proteins (R: *n* = 5, NonR: *n* = 3) from patients with prior topical treatment. C&RT identified IL-6 and CXCL8 as classifiers (accuracy = 100%, sensitivity = 100%, specificity = 100%); (**D**) Classification tree model for anti-IL-23 response (SR: *n* = 4, NonR: *n* = 7) based on five hub proteins from patients with prior biologic treatment. C&RT identified CCL20, IL-6, and CXCL8 as classifiers (accuracy = 90%, sensitivity = 100%, specificity = 85%).

**Table 1 ijms-26-08118-t001:** Characteristics of psoriatic patients.

	Patients	Anti-IL-23 (*n* = 18)	Anti-TNFα (*n* = 16)	*p*-Value
Gender Male/Female	23/11	13/5	10/6	n.s
Age at baseline	51.00 (36.50–56.50)	53.00 (37.00–65.00)	49.00 (35.50–53.50)	n.s
Age at diagnosis	32.00 (17.75–52.50)	39.00 (28.50–64.50)	17.50 (11.25–34.25)	0.02
PASI Baseline	10.80 (6.95–14.95)	11.70 (9.80–14.83)	6.95 (4.90–10.75)	n.s
BSA Baseline	14.20 (8.85–27.15)	16.25 (9.75–25.75)	11.55 (7.52–23.20)	n.s
PGA Baseline				0.01
2	3	0	3	
3	12	3	9	
4	10	9	1	
5	9	6	3	
Previous treatments				n.s
Topical	11	3	8	
Apremilast	3	1	2	
Anti-TNFα	4	4	0	
Anti-IL-17	2	2	0	
Anti-p40	5	5	0	
Anti-p19	1	1	0	
Methotrexate	6	2	4	
Cyclosporine	2	0	2	
Fumarate	1	0	1	

n.s not significant.

**Table 2 ijms-26-08118-t002:** Characteristics of psoriatic patients based on previous treatment.

	Topical (*n* = 10)	Biologic (*n* = 11)	Other (*n* = 12)	*p*-Value
Gender M/F	7/3	6/5	8/4	n.s
Age at baseline	48.50 (37.00–54.25)	60.00 (46.00–75.00)	48.00 (35.25–53.00)	n.s
Age at diagnosis	24.00 (15.00–47.00)	51.00 (29.00–64.00)	25.00 (12.25–35.75)	0.056
PASI Baseline	12.85 (5.92–25.25)	10.80 (8.40–12.60)	7.95 (6.20–12.50)	n.s
BSA Baseline	14.70 (10.00–57.50)	13.00 (8.50–17.00)	15.80 (7.52–30.25)	n.s
PGA Baseline				n.s
2	1	0	2	
3	3	3	6	
4	2	6	1	
5	4	2	3	

n.s not significant.

**Table 3 ijms-26-08118-t003:** Characteristics of psoriatic patients based on response classification to anti-TNFα treatment.

	Anti-TNFα		*p*-Value
	R	NonR	
Gender M/F	2/5	7/2	n.s
Age at baseline	52.50 (35.50–69.00)	48.00 (31.75–51.75)	n.s
Age at diagnosis	22.50 (12.75–48.25)	17.50 (9.75–41.00)	n.s
PASI Baseline	11.00 (3.80–28.00)	6.50 (6.25–7.95)	n.s
BSA Baseline	24.60 (6.30–80.00)	10.50 (7.95–14.70)	n.s
PGA Baseline			0.08
2	2	1	
3	1	7	
4	1	0	
5	3	1	
Previous treatments			n.s
Topical	5	3	
Apremilast	0	2	
Anti-TNFα	0	0	
Anti-IL-17	0	0	
Anti-p40	0	0	
Methotrexate	1	2	
Cyclosporine	0	2	
Fumarate	1	0	

n.s not significant.

**Table 4 ijms-26-08118-t004:** Characteristics of psoriatic patients based on response classification to anti-IL-23 treatment.

	Anti-IL-23		*p*-Value
	SR	NonR	
Gender M/F	7/1	5/4	n.s
Age at baseline	52.00 (37.25–73.25)	62.00 (41.50–73.00)	n.s
Age at diagnosis	31.50 (19.00–58.75)	51.00 (32.00–61.00)	n.s
PASI Baseline	11.80 (8.55–16.53)	11.70 (10.00–14.95)	n.s
BSA Baseline	19.75 (9.25–31.75)	16.00 (9.25–20.50)	n.s
PGA Baseline			n.s
2	0	0	
3	2	1	
4	2	6	
5	4	2	
Previous treatments			n.s
Topical	2	1	
Apremilast	1	0	
Anti-TNFα	2	2	
Anti-IL-17	0	2	
Anti-p40	2	3	
Methotrexate	1	1	
Cyclosporine	0	0	
Fumarate	0	0	

n.s not significant.

## Data Availability

The sequence data generated in this study have been deposited in NCBI Gene expression Omnibus (GEO/SRA) under the accession number BioProject ID PRJNA1291610.

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
