# Peer review of "Transcriptomic Identification of Immune-Related Hubs as Candidate Predictor Biomarkers of Therapeutic Response in Psoriasis"

_ijms, 2025, doi:10.3390/ijms26178118_

Round 1
Reviewer 1 Report
Comments and Suggestions for Authors
1. the correlation between plasma biomarkers (such as CCL20) and skin gene expression is weak (only CCL20 in skin lesions r=0.351, p=0.045), but the conclusion emphasizes its value as a predictive marker, and the evidence is insufficient (Figure 6D).
2. 66.7% of patients in the anti-il-23 group had a history of biological agent treatment, while only 25% in the anti TNF α group (Table 1). The type of previous treatment (such as anti TNF α, anti-IL-17) may affect the baseline immune status, but it was not included in the efficacy analysis as a covariate.
3. the definition of anti TNF α efficacy as "pasi=0 at 6 months" is too strict (pasi75/90 is commonly used in clinic), which may lead to a high proportion of "non responders" (9 / 16). The anti-il-23 group used "pasi=0 at 3 and 6 months" to define "super responders", but did not specify the basis for selecting this standard.
4. RNA SEQ analysis does not mention batch effect correction
5. key illustrations such as figure 1E and figure 2a are not fully interpreted in the text (such as the meaning of "red node").
6. the clinical application of skin biopsy (invasive) as a predictive marker is limited, and the predictive value of plasma markers is not fully verified.
Reviewer 2 Report
Comments and Suggestions for Authors
Dear Authors,
Your paper has scientific potential, but some corrections must be applied:
Experimental Design: Please provide more detail on sequencing depth (reads per sample) and any batch/randomization procedures. As Conesa et al. note, choosing appropriate sequencing depth and replicates is crucial. Was sample processing randomized across batches to avoid confounding? Include sample counts and a statement on any technical replicates or quality control (e.g. % mapped reads).
Controls and Cohort: Consider adding or discussing healthy control skin for comparison. Nonlesional skin is a reasonable internal control, but differences with truly healthy skin could contextualize your DEGs (Krueger et al., 2009). Also, clarify inclusion criteria and any power analysis. With 34 patients divided into subgroups (e.g. only 7 TNF-responders), statistical power is limited. A power calculation would help justify the sample size.
Differential Expression Analysis: Clarify normalization and multiple-testing correction. You mention limma-voom but not TMM or other normalization before voom; specify the approach. Confirm that “adjusted p-value” refers to FDR (e.g. Benjamini-Hochberg). The fold-change cutoff (|log₂FC|≥1.5) is stringent; justify this choice. For the topical vs biologics pre-treatment comparison, the criteria (FC ≥1 or ≤1) seem to include all genes (since FC of 1 is no change) – please clarify (perhaps you meant log₂FC ≥1).
GO and Network Analysis: The use of ShinyGO and ClueGO is fine, but consider citing or explaining the rationale for parameter choices (e.g. why STRING score >0.4, why MCC algorithm in CytoHubba). Provide the number of input genes for each step. It may help to report any validation of hub selection (e.g. consistency across algorithms). Many studies (e.g. Su et al., 2021; Li et al., 2023) have identified psoriasis hub genes – please compare your hubs with such lists in the discussion.
Statistical Analysis: The statistical methods list many tests (t-test, ANOVA, Mann–Whitney, etc.). Ensure that the assumptions (normality) were checked. For multiple group comparisons, clarify how p-values were adjusted. The correlation matrices and heatmaps (Fig. 2C) are useful; include p-values or significance levels for key correlations in the text or figure.
Decision Tree Models: This is a central claim, so please detail the method. Which variables (genes, cytokines) were input to the trees? How were “responders” defined (PASI=0 seems strict; is this clinically standard)? How were tree models validated? The use of C&RT in XLSTAT/GraphPad is unconventional for reproducibility – consider re-running with an open-source package (e.g. R’s rpart with cross-validation). At minimum, report confusion matrices or cross-validated accuracy to demonstrate the model’s robustness. The current 100% purity suggests overfitting on a small dataset. Also, be explicit about whether expression values were log-transformed (they mention log CPM), and provide units (e.g. for plasma cytokine pg/mL vs log cpm).
Biomarker Claims: The term “biomarker” and “predictor” implies potential clinical use. Emphasize that these findings are exploratory. It would be more accurate to say “candidate biomarkers” and note that validation (e.g. qPCR, independent cohort) is needed. Other authors validated hub genes by qPCR/IHC; without such validation, conclusions should be cautious. The Discussion partially acknowledges this (“Future research should validate these biomarkers”), which is good, but consider adding a sentence in the conclusion about the need for prospective validation.
Results Clarity: Improve figure and table quality. Ensure all figures have clear legends and labeled axes. For example, in decision-tree figures, label the cutoff values and sample counts at each node. In heatmaps, include color scales. Table 3 (patient response characteristics) needs clearer formatting (perhaps split into two tables for TNF vs IL-23 responses). When reporting statistical differences, include exact p-values (or state “p<0.05”) and clarify what comparisons were tested.
Discussion and References: The Discussion nicely relates findings to other studies. However, it could more critically address limitations: e.g. lack of healthy controls, small N, single time-point for RNA-seq, and potential confounders (age, previous treatments). Also, discuss any surprising findings (e.g. IL-1β not being among top hubs, or why CXCL13 was included). You might cite Ainali et al. 2012 to support the use of tree methods, and mention that random forests often outperform single trees for high-dimensional data.
Data Availability: The manuscript says data are available on request. IJMS typically expects public deposition of RNA-seq data (e.g. GEO/SRA). Please consider uploading raw sequence data and processed counts to a public repository for reproducibility. This aligns with best practices
Comments on the Quality of English LanguageThe English language is generally clear and understandable, with appropriate scientific terminology. However, the manuscript would benefit from careful proofreading to improve flow, correct minor grammatical issues, and clarify dense or awkward phrasing—especially in the Results and Discussion sections. Some figure legends and methodological descriptions are overly long or complex. A professional language edit is recommended to enhance readability and precision.
Round 2
Reviewer 1 Report
Comments and Suggestions for Authors
The author has addressed my concerns. Thank you to the author for their efforts in revising the manuscript.
Author Response
Thank you for your positive feedback. I appreciate your careful review and am pleased that our revisions have addressed your comments satisfactorily.
Reviewer 2 Report
Comments and Suggestions for Authors
Dear Authors,
I commend you for the thoroughness of the revision. The added methodological detail (40–50 million reads per sample, randomized processing, quality metrics, normalization and FDR correction) and the clear explanation of fold‐change thresholds make the study more transparent and reproducible. The expanded discussion of non-lesional versus truly healthy skin and the explicit statement of the exploratory, hypothesis-generating nature of the findings provide important context. Justifying your bioinformatic parameter choices (STRING score >0.4, the MCC hub algorithm) and identifying how many genes were analyzed in each step helps the reader follow your analysis. The figures are now clearer – with labeled axes, color keys, and exact p-values – and the decision-tree models appropriately include input features and performance metrics. It is also good to see that you have adopted the term “candidate biomarkers” and cited the need for independent validation.
A few minor points remain: ensure that the final manuscript includes the GEO/SRA accession numbers once the data are deposited, and perform a final proofread to catch any remaining language or formatting issues. Overall, the extensive revisions have significantly improved the clarity and rigor of the paper. The manuscript is now much stronger and should be of interest to the field.
Comments on the Quality of English LanguageThe manuscript is generally written in clear, comprehensible English, but it still needs a final language polish. A careful proof-read (or a light professional copy-edit) should eliminate several small issues that remain: scattered typographical slips (e.g., “cwould,” “Sadistically differences,” “struggesting”), occasional missing articles and prepositions (“in accordance with clinical guidelines,” “aims to address”), inconsistent verb tenses in the Results section, and a number of very long sentences that would read more smoothly if broken in two. Abbreviations and hyphenation should be standardized throughout (e.g., “anti-TNFα,” “IL-23,” “RNA-seq”), and a few figure legends could be trimmed to remove redundancy. Addressing these minor points will further improve readability and professionalism.
Round 3
Reviewer 2 Report
Comments and Suggestions for Authors
Dear Authors,
The study is timely and well-conceived, integrating transcriptomic and plasma data to identify immune-related biomarkers of therapeutic response in psoriasis. Your revisions have substantially improved clarity and addressed prior reviewer concerns, particularly regarding data availability, language consistency, and methodological transparency.
To further enhance the manuscript, we recommend a careful final proofreading—especially of figure legends, where typographical errors and redundancies persist. Ensure uniform formatting in tables and correct small phrasing issues (e.g., “according toas per”). Consider adding a brief concluding paragraph or section summarizing the main findings and clinical relevance.
These minor refinements will strengthen the presentation of an otherwise solid and relevant study.
Comments on the Quality of English LanguageThe quality of English in the manuscript is generally acceptable and substantially improved from the previous version. However, several typographical errors, grammatical inconsistencies, and awkward phrasings remain—particularly in the figure legends and select sections of the main text. A final careful proofreading or light professional language editing is recommended to ensure clarity and polish throughout.
